# SSRIs and SNRIs (SRI) in Pregnancy: Effects on the Course of Pregnancy and the Offspring: How Far Are We from Having All the Answers?

**DOI:** 10.3390/ijms20102370

**Published:** 2019-05-14

**Authors:** Asher Ornoy, Gideon Koren

**Affiliations:** 1Laboratory of Teratology, Department of Medical Neurobiology, Hebrew University Hadassah Medical School, Jerusalem 91120, Israel; 2Maccabi Institute of Research and Innovation, Tel Aviv University, Tel Aviv, Israel and Ariel University, Ariel 40700, Israel; gidiup_2000@yahoo.com

**Keywords:** SRI, SSRIs, SNRIs, pregnancy, malformations, neurodevelopmental effects, ASD, epigenetic effects

## Abstract

Serotonin has important roles in the development of the brain and other organs. Manipulations of synaptic serotonin by drugs such as serotonin reuptake inhibitors (SRI) or serotonin norepinephrine reuptake inhibitors (SNRI) might alter their development and function. Of interest, most studies on the outcome of prenatal exposure to SRI in human have not found significant embryonic or fetal damage, except for a possible, slight increase in cardiac malformations. In up to a third of newborns exposed to SRI, exposure may induce transient neonatal behavioral changes (poor neonatal adaptation) and increased rate of persistent pulmonary hypertension. Prenatal SRI may also cause slight motor delay and language impairment but these are transient. The data on the possible association of prenatal SRIs with autism spectrum disorder (ASD) are inconsistent, and seem to be related to pre-pregnancy treatment or to maternal depression. Prenatal SRIs also appear to affect the hypothalamic hypophyseal adrenal (HPA) axis inducing epigenetic changes, but the long-term consequences of these effects on humans are as yet unknown. SRIs are metabolized in the liver by several cytochrome P450 (CYP) enzymes. Faster metabolism of most SRIs in late pregnancy leads to lower maternal concentrations, and thus potentially to decreased efficacy which is more prominent in women that are rapid metabolizers. Studies suggest that the serotonin transporter SLC6A4 promoter is associated with adverse neonatal outcomes after SRI exposure. Since maternal depression may adversely affect the child’s development, one has to consider the risk of SRI discontinuation on the fetus and the child. As with any drug treatment in pregnancy, the benefits to the mother should be considered versus the possible hazards to the developing embryo/fetus.

## 1. Introduction

Serotonin (5 hydroxy tryptamine: 5-HT) is derived from the amino acid tryptophan. Serotonergic neurons are restricted to brain stem raphe nuclei projecting in many areas of the cerebral hemispheres [1,2], including the prefrontal cortex, hippocampus, nucleus accumbens, corpus striatum [1,2]; in periphery, serotonin is synthesized by the entero-chromaffin cells of the gastrointestinal system and stored in platelets [1,2].

Serotonin is a mood and emotional regulator, regulator of motor activity, memory, cognition, sleep and appetite. Serotonin has several important roles during embryonic and fetal brain development including neuronal maturation, migration, synaptogenesis and differentiation of neural crest cells which are involved in facial and cardiac development [1,3,4]. It is also an important factor in several epigenetic processes such as stress responsivity via the hypothalamic–pituitary–adrenal axis [3,4]. 

It is logical to expect that drugs which affect serotonin metabolism such as tricyclic and tetracyclic antidepressants, selective serotonin reuptake inhibitors (SSRIs) and selective serotonin norepinephrine reuptake inhibitors (SNRIs) would be able to affect embryonic and fetal brain development and induce various neurobehavioral and other deficits. By crossing the fetal blood–brain barrier, they may increase brain serotonin and modify serotonin signaling, potentially altering behavior at childhood, adolescence and adulthood [3].

The use of SRI during pregnancy is steadily increasing, reaching in recent years 2–6% of all pregnancies, with recent average figures of 3.4% [5,6,7]. They are prescribed for the treatment of depression, anxiety, obsessive compulsive disorder (OCD), panic disorder and phobias. Importantly, many animal experiments showed adverse effects on pregnancy outcome when SRI are used inducing anatomical and neurobehavioral deficits [3,8]. In spite of the fact that SRI appear to be the most studied drugs during pregnancy, there are still conflicting views on the risks of these drugs on the course of pregnancy [5,9,10]. 

SRIs inhibit the serotonin transporter and increase serotonin concentration in the synapses. Serotonin transporter was also isolated in the human placenta [11]. SRI were shown to cross the human placenta [12] and are excreted in relatively small amounts in human milk. In spite of most extensive studies on the effects of SRI in pregnancy, there are still conflicting views on their possible adverse effects on the course of human pregnancy and on the newborn infant. It is therefore important to weight the benefits of SRI treatment to the mother against the possible hazards to pregnancy outcome before making decisions on treatment.

The aim of the present review is to evaluate the possible risks of SRI use during human pregnancy including congenital malformations, pregnancy complications, poor postnatal adaptation and possible long-term neurodevelopmental and psychiatric effects. We will also discuss pharmacokinetic and pharmacodynamic changes during pregnancy and strive to reach conclusions based on the available extensive published literature, while comparing to data controlling for disease state and the consideration of confounding factors. We will discuss SRI and SNRI as a group because discussion of individual drugs is beyond the scope of this review. However, one should be aware that the many drugs of these groups may differ in their pharmacological and metabolic characteristics. Neither will we discuss animal data. Most available literature relates to SRIs, as they have been used more often than SNRIs.

## 2. Major Congenital Malformations and Cardiac Anomalies 

Over the last two decades numerous cohort and case control studies have investigated possible associations between gestational exposure to SRI and the risk of major malformations (see Table 1). No other class of medications has been studied so frequently and yet, despite very large numbers, there is still controversy whether the SRI are associated with increased teratogenic risk in general, and mostly cardiac malformations. So far, any specific syndrome or type of malformation related to any of the SRI has not been absolutely delineated [9,10,13]. However, a possible association between the use of SRI in pregnancy and increased rate of cardiac anomalies as well as some increase in several rare major malformations was described in many studies repeatedly [14,15,16,17,18,19,20,21]. The cardiac anomalies reported were generally ventricular septal defects, right ventricular outflow tract obstruction and recently also severe cardiac malformations [15,20,21].

Inconsistencies among studies exist in the definition of malformations of the heart, some including small atrial and ventricular septal defects that often close spontaneously, while others excluding such anomalies. Other inconsistencies among studies are differences in study design, control of smoking, alcohol drinking, additional use of other medications or maternal underlying diseases. Daud et al. [22] in their meta-analysis on SRI and cardiac malformations hypothesized that the contradictory results may be related to genetic differences among populations studied and other confounding factors.

The earliest description of an increase in the rate of cardiac malformations (Table 1) were related to paroxetine, with several studies demonstrating an increased rate of anomalies, mainly ventricular septal defect (VSD) [5,15] and atrial septal defects (ASD) [23]. These anomalies were shown to be related only to first trimester paroxetine treatment and were described as dose dependent [24]. In a prospective cohort study published by us with several European Teratogen Information Services [15], the increased rate of cardiac malformations was described following treatment with both paroxetine (adjusted Odds Ratio (aOR) 2.66, 95% CI 0.80,8.90) and fluoxetine (aOR 4.47, 95% CI 1.31, 15.27). However, when adjusting for confounding factors including maternal smoking, only exposure to fluoxetine remained significant. Merlob et al. [25] carried out echocardiograms on any newborn infant with a persistent cardiac murmur and found an increased rate of cardiac malformations following prenatal exposure to different SRI. In this study the overall rate of cardiac malformations in the SRI exposed infants was twice that observed in non-exposed infants (3.4% versus 1.6%). 

Knudsen et al. [21] performed a cohort study using the EUROCAT definitions of malformations with citalopram being the most commonly prescribed drug (34.5%). They found a similar rate of atrial and of ventricular septal defects in the exposed and non-exposed children, but the rate of “severe” cardiac malformations was four times higher compared to non-exposed (aOR 4.03; CI 1.75–9.26). Jordan et al. [20] evaluated the outcome of deliveries from EUROCAT registries regarding congenital malformations in women who were prescribed SRI. They also found an increase in the rate of “severe” cardiac malformations (OR of 1.59; CI 1.06–2.11) and a dose-dependent relation.

In another study [26] conducted on 12 EUROCAT congenital anomaly registries of 2.1 million births, there were 12,876 infants and fetuses with cardiac malformations. A higher rate of cardiac malformations was found among the offspring of mothers using SRI (aOR 1.41, 95% CI 1.07–1.86) including paroxetine (aOR 1.43) and fluoxetine (aOR 1.56). There was also an increase in severe cardiac malformations such as Tetralogy of Fallot (aOR 3.16) and Ebstein’s anomaly (aOR 8.23). The rate of several other anomalies was increased as well (e.g., gastroschisis, ano/rectal stenosis, clubfoot).

These large studies are clearly suggestive of an increased rate of cardiac anomalies. However, one should remember the limitations of registries and data based on prescriptions without accurate knowledge, as women may not have actually taken the drugs [27]. In addition, there is a lack of data on the possible effects of the maternal disease, confounding by indication. 

There are other cohort studies where no increase in cardiac or other congenital malformations were observed compared with non-exposed children born to mothers with or without depression [28,29,30,31,32,33,34]. 

Jimenes–Solem et al. [28] performed a large national population-based study on the relation between SRI use during pregnancy and major congenital anomalies, especially of the heart, and compared their rate to that of children born to women with depression that did not take SRI during pregnancy. They found that the rate of cardiac malformations was similar in the offspring of women who took SRI and those who stopped SRI treatment before pregnancy, implying that the increase in cardiac anomalies was apparently related to maternal disease, by increasing the likelihood of monitoring fetal and neonatal heart over that of healthy control women [28]. Indeed, such a detection bias has been shown in a Canadian study [35]. Byatt et al. [29] also found that women with depression and anxiety were more likely to attend medical care with their infants when compared to healthy women. This increases the rate of detection of cardiac murmurs and their follow up [29]. 

There have been numerous systematic reviews reporting potential connection between SRI use in pregnancy and cardiac malformations. In the meta-analysis published by Bar-Oz et al. [35] comparing the outcome women using paroxetine to non-exposed pregnancies the authors found that first trimester paroxetine exposure was associated with a significant increase in the risk of cardiac malformations, with an overall odds ratio of 1.72. The women using antidepressants during pregnancy performed more cardiac examinations in their children that might explain at least some of the results.

Wurst et al. [36] found an increased risk for combined cardiac defects (aOR 1.46) and aggregated congenital defects (aOR 1.24) with first trimester paroxetine use. Several opposing commentaries on this meta-analysis, stating that the meta-analysis did not include several important negative studies, were published thereafter [37].

Gao et al., in a recent review [38] of 29 cohort studies with 59,894 SRI treated women, found a slight increase in the rate of congenital malformations (RR 1.11, 95% CI 1.03–1.19) and a more prominent increase in the rate of cardiac malformations (RR 1.24, 95% CI 1.11–1.37). The highest increase was of cardiac septal defects ASD and VSD. They concluded that the risk for congenital cardiac and other malformations due to serotonin exposure in the first trimester of pregnancy is low.

There are also several “negative” meta-analyses who reported that there is no increase in the rate of cardiac anomalies after prenatal exposure to SRI [39,40,41,42,43,44]. Obrien et al. [39] did not find increased cardiovascular anomalies in children prenatally exposed to paroxetine. Similar results were described by Addis and Koren [40] and by Einarson and Einarson [41]. Rahimi et al. [42] found in children prenatally exposed to SRI a slight increase in spontaneous abortions but not in major or cardiac malformations. 

In summary: the current data on pregnancy outcome in women using SRI in pregnancy do not fulfil the criteria for proof of teratogenicity as outlined by Shepard [45] and by Brent [46]. The epidemiologic studies are inconclusive, there is no dose response relationship and there is no defined syndrome. Moreover, studies with disease controls were mostly negative.

The overwhelming message of recent meta-analyses is of a small but significant increase in risk of cardiovascular malformations, especially VSD and some severe cardiac malformations. However, most studies included in these meta-analyses did not adjust for the important ascertainment bias inherent in their design, as follows:

Women treated with SRI-SNRI are more likely to have their fetuses-neonates examined for cardiovascular malformations than control women. Because many of these malformations consist of VSD, which are mostly muscular and tend to close spontaneously during infancy, they are substantially more likely to be missed in the control groups of untreated women [35]. This concept has also been supported by Jimmenez–Solem et al. [28] who demonstrated, as stated above, that depressed women are twice more likely than healthy women to have children with congenital heart malformations, either when treated with SSRI-SNRI, or when they avoid such treatment [28]. This inherent bias means that the combination of numerous studies, all biased in the same direction, is not likely to help discern teratogenic risk. However, it might be advisable to offer fetal echocardiography for prenatal detection of severe cardiac malformations.

Because depression in late pregnancy is a major predictor for the life threatening postpartum depression, discontinuing antidepressants in pregnant women can be life threatening. This risk must be balanced against the tendency to discontinue SRI among pregnant women due to perceived, but still unproven teratogenic risk. We should, however, consider the possibility that SRI have other adverse effects on the developing fetus, possibly affecting fetal growth, inducing early delivery and difficulties in postnatal adaptation [9,10].

## 3. Complications of Pregnancy 

The effects of SRI on pregnancy outcome include spontaneous abortions, intrauterine fetal death, preterm delivery and intrauterine growth restriction (IUGR, FGR). In general, in studies controlling for maternal illness and other confounders, there was no significant detrimental effect of SRI on any of these outcomes.

### 3.1. Spontaneous Abortions, Fetal and Perinatal Death 

SRI and other antidepressants have been associated with an increased rate of spontaneous abortions and stillbirths as described by Ban et al. [47] who found a relative risk for spontaneous abortions of 1.5 and 1.6 for perinatal death (see Table 1). Several additional prospective cohort studies concurred in showing a slight increase in the rate of spontaneous abortions [15,48,49]. In our prospective study [15] the single predictor for spontaneous abortion was the gestational age at initial contact, as the earlier the contact, the higher is the rate of miscarriage since the majority of miscarriage occur early in pregnancy. A similar finding was described by Johansen et al. [50] reporting 10.7% of spontaneous abortions in women enrolled in their follow up at their first antenatal care visit compared to only 4% in women enrolled later during their pregnancy. Moreover, in women who took SRI but not for depression or anxiety, there was no increase in the rate of miscarriages demonstrating that depression might be the cause and not SRI [50]. Unfortunately, in most studies there was no control for the underlying maternal illness. Similarly, Andersen et al. [51] found that the rate of miscarriages was 12.6% compared to 11.1% in non-exposed. However, women who stopped taking SRI before becoming pregnant also had a high rate of early miscarriages. In a recent study Richardson et al. [52] compared the rate of spontaneous abortion in women treated during early pregnancy with venlafaxine—an SNRI—with SRI and unexposed women and found, after correction for the gestational age at reporting, no differences in the rate of miscarriage between the three groups.

One may conclude that SRI are not associated with increased rates of miscarriages, and the slight increase in early miscarriages is connected to the time of pregnancy at the first contact and/or with the underlying maternal disease.

### 3.2. Prematurity and Intrauterine Growth Restriction (IUGR) 

The risks of small for gestational age (SGA) and prematurity were increased in infants following prenatal exposure to SRI in spite of control for maternal illness [49,53,54] (Table 1). Oberlander et al. [18] found that the rate of prematurity was doubled among women taking SRI compared to women with or without a history of psychiatric morbidity. Some studies reported that the risk of prematurity was not increased among SRI users, while the risk of SGA offspring did increase [15,52]. Ramos et al. (55] and Kalderon et al. [56] found an increased risk for prematurity only after taking SRI in the second or third trimesters of pregnancy [55,56].

In a study by Toh et al. [57], no association was found between SRI and the risk of SGA, regardless of the time of exposure. In contrast, Malm et al. [58] found that untreated and SRI treated women with psychiatric disease have a higher rate of preterm birth compared to control women, but SRI lower that rate in comparison to the untreated women. El Marroun et al. found [59] in a prospective cohort study that the use of SRI in pregnancy was not associated with IUGR when compared to offspring of untreated depressed mothers implying the importance of the underlying maternal disease. In many studies this confounder was not adjusted for (see Table 1). 

A recent meta-analysis of 15 studies with almost two million cases, has suggested a 45% increase in risk of suboptimal fetal growth. However, these studies did not adjust for maternal lifestyle, illness, time of exposure in pregnancy and the specific medications used [60].

In summary: most of the studies reporting an increased risk of IUGR or preterm delivery are potentially confounded by the underlying psychiatric disorder because women with psychiatric disorders seem to have a higher risk for preterm delivery and IUGR [61,62].

## 4. Neonatal Effects 

These are a variety of adverse “withdrawal symptoms” which may appear during the first few postnatal days in babies prenatally exposed to SRI. Most of them are transient and have no long-term negative effects (see Table 2).

### 4.1. Poor Neonatal Adaptation

Prenatal exposure to psychotropic drugs, including SRI, especially during the last months of pregnancy, may cause poor neonatal adaptation syndrome in up to 30% of infants [63,64,65,66,67]. Clinical findings include: irritability, abnormal crying, tremor, jitteriness, lethargy, respiratory distress, poor muscle tone, and, rarely, convulsions.

Norby et al. [67] found that newborn infants prenatally exposed to SRI late in pregnancy had about two thirds more admissions to neonatal intensive care units compared to non-exposed. In children exposed only early in pregnancy the rate of admission was almost normal.

In summary: it is imperative for the clinician to be aware of possible poor neonatal adaptation symptoms in newborn infants born to mothers using psychiatric drugs, especially SRI, late in pregnancy.

### 4.2. Persistent Pulmonary Hypertension of the Newborn (PPHN) 

The association between prenatal exposure to SRI late in pregnancy and PPHN has been described by many investigators [6,68,69,70,71,72]). The absolute risk is less than 1% and the clinical course is less severe compared to typical PPHN resulting from other causes (i.e., diaphragmatic hernia, aspiration). PPHN in offspring prenatally exposed to SRI was also associated with caesarean delivery more than to SRI use [71]. Huybrechts et al. found [6] that the rate of PPHN is also high in the offspring of women with depression who did not use SRI, and the additional risk imposed by SRI is small (see Table 2).

In a recent meta-analysis of 11 studies reporting on a total of 156,978 women and their offspring exposed to SRI [73] the risk of PPHN was increased by 82% (OR 1.82, 95% confidence interval 1.31–2.54). Similar figures emerged when restricting the analysis to exposure after 20 weeks of gestation.

In summary: the existing studies show that prenatal SRI exposure is associated with a slight increase in PPHN and poor neonatal adaptation which may appear even several days after birth. The exposed newborn should be carefully monitored for any medical problem developing in the first few days after delivery to diagnose withdrawal symptoms or persistent pulmonary hypertension [66]. The present practice of rapid release of babies after birth may not be appropriate for children of depressed mothers because of these symptoms, coupled with the increased risk for maternal postpartum depression.

## 5. Neurodevelopmental Effects

SRI cross the fetal blood–brain barrier and might, theoretically, affect brain function as shown by transient nervous system changes manifested by perinatal adaptation difficulties [65,66]. However, the withdrawal effects are temporary and do not appear to have long-term clinical significance [74]. Still, one has to rule out long term neurobehavioral and psychiatric effects that are not diagnosed during the neonatal or perinatal period, as they are manifested only later at childhood or even during adulthood. In addition, the environment where the child was raised in his early years should also be considered, as subtle prenatal brain damage might be minimized if the child is raised in a good home environment [75,76]. We have previously shown that in children of heroin addicts the cognitive achievements largely depend on early life environment, and in adopted children prenatally exposed to heroin raised in a good environment the cognitive ability in early childhood was normal, while unexposed children raised in an unfavorable environment have neurodevelopmental delay [76,77]. However, in order to study long term development, children must be followed up to adulthood and investigators should take into account the possible effects of maternal illness. Long term studies are missing for most psychotropic drugs as most studies were carried out on children and this may have limited prediction for neurobehavioral deviations at adolescence or adulthood, especially for psychiatric diseases [78].

There are many animal studies, especially in rodents, showing adverse effects of prenatal or neonatal SRI on postnatal development [8,78]. The human data, however, is generally reassuring. Since the neurodevelopmental outcome is not the main purpose of this review, we will summarize this issue only briefly. Nulman et al. in several studies [79,80,81] compared the developmental outcome of children prenatally exposed to fluoxetine, children exposed to venlafaxine, children exposed to other antidepressants to control children and found no differences among the four groups in their developmental outcome. They also compared the development of sibling-one prenatally exposed to SRI and the other not exposed and found no difference between them. The siblings were raised in the same home, thus correcting for the possible effects of early environment on their development [82].

Several other investigators [83,84] found no developmental differences between SRI-exposed and unexposed infants but found abnormal scores in internalizing or externalizing behaviors were observed in children born to mothers with psychiatric problems unrelated to SRI exposure [83,84]. These are expected, as mother-child relationships and child’s behavior are significantly affected by maternal psychiatric morbidity. 

Other investigators have assessed the development of SRI exposed children who exhibited withdrawal symptoms (NAS) in comparison to those that had normal neonatal adaptation. Klinger et al., for example, found no difference between the groups, all developing within the normal range [74].

There are several studies demonstrating slight, but transient, motor and language delay. For example: Casper et al. [85] found no differences in mental developmental achievements on the Bayley scales of infant development between SRI-exposed and unexposed infants but found lower indexes on the psychomotor scales in the SRI exposed children. Similarly, Pedersen et al. [86] found that second or third trimester exposure to antidepressants was associated with a transient delay in gross motor developmental milestones, with catch up following increased children’s age. Similar results were found by Hanley et al. [87] and Santucci et al. [88] who reported that infants born to mothers treated with SRI had lower gross motor abilities that generally persisted only in the first year. Johnson et al. [89] studied prenatally SRI-exposed young children and found normal cognitive abilities but a slight delay in expressive language and behavioral abnormalities.

Brown et al. [90] evaluating 845,345 singleton births studied the development of children born to mothers that received at least two prescriptions of SRI during pregnancy. They investigated the neurodevelopmental outcome of 15,596 exposed children compared to 9537 unexposed children. The SRI exposed children had low scores in their motor, language and scholastic abilities. Since they used hospital discharge records and not well defined psychometric evaluation, the validity of these data is questionable. Malm et al. [91] used the same population and observed in the SRI exposed children a high rate of depression during adolescence.

Recently Kragholm et al. [92] studied the performance of 3314 elementary school children prenatally exposed to SRI compared to 3536 unexposed children attending the same school. They found that the SRI exposed children started school at a later age but there were no specific learning problems attributed to SRI.

In summary: it seems that prenatal exposure to SRI may induce some transient motor delay. However, there seems to be no evidence for long-term adverse developmental effects. As most studies were carried out on young children, late adverse neurodevelopmental effects on learning, attention span and increased rate of mental illness cannot be completely excluded.

## 6. Possible Association with Autism Spectrum Disorder (ASD)

Children with ASD often have elevated serotonin in the platelets, raising the possibility of an association between prenatal exposure to SRI and ASD [93]. Not only is this issue in debate, but there is no agreement among the studies during which period in pregnancy is the highest susceptibility to ASD.

Croen et al. [94] reported that the risk of ASD following first trimester exposure to SRI is doubled. They found that 6.7% of mothers of children with ASD used antidepressants compared to only 3.3% in the controls. This case control study was based on prescriptions of SRI.

Harrington et al. [95] studied possible associations of ASD and prenatal SRI in 492 ASD children compared to 154 children with neurodevelopmental delay and 320 normal children, and reported a significant association in boys, especially if exposed during the third trimester of pregnancy.

Boukhris et al. [96] identified in the Quebec pregnancy and children cohort 1054 (0.73%) children with ASD among over 145,000 full term singleton infants. Prenatal exposure to SRI, especially in the second and third trimester of pregnancy was associated with a higher risk for ASD even after controlling for maternal depression. 

In contrast to these studies suggesting an association of SRI with ASD, there are relatively large studies that did not find such associations. For example: two studies from the US [97,98] on a total of 2622 children with ASD found no association of prenatal exposure to SRI and ASD.

Rai et al. [99] in a population-based case control study found that more mothers of children with ASD had depression (1%) as opposed to only 0.6% in mothers of children without ASD [99]. The use of antidepressants in the mothers with depression additionally elevated the risk for high functioning ASD. However, such use may reflect more severe forms of depression, rather than an independent effect of the SRI. The Avon longitudinal study of parents and children birth cohort [100] also found that ASD is associated with maternal psychosis. 

Three other population-based studies [101,102,103] found no association between prenatal SRI exposure and ASD. Of notice is the large population-based study by Sujan et al. [103] on 22,544 infants that found no association between first trimester use of antidepressant and ASD in the offspring. These studies suggest that when controlling for confounders, the association between prenatal SRI and ASD disappears.

Malm et al. [91] also reported no association of prenatal exposure to SRI and ASD in the offspring. They found, however, an increased rate of psychiatric disorders among exposed children at adolescence.

A rather convincing demonstration that parental psychiatric disorder is apparently the underlying cause for the possible association between prenatal SRI and ASD is the relatively recent study published by Yang et al. [104] who found that paternal use of SRI during the last 3 months before conception was linked to an increased rate of ASD in the child (RR 1.62, (95% CI 1.33–1.96)). 

Published reviews and meta-analyses related to the possible effects of SRI on ASD also came to conflicting conclusions. Man et al. [105] reviewed 12 studies, and concluded that there is support for an increased risk of ASD following prenatal SRI exposure. Andalib et al. [106] evaluated all studies in English that looked at the possible association between prenatal exposure to SRI and ASD and found a significant association with an OR of 1.82 (95% CI 1.59–2.10). Gentile [107] in his review of the English literature reported on some association but could not reach any definite conclusions because of many unconsidered confounding factors. Kaplan et al. [108] found in a meta-analysis an association between pre-pregnancy treatment with SRI and ASD but not with treatment during pregnancy. Similarly, Mezzacappa et al. [109] concluded that the strongest association between maternal SRI and ASD was found with treatment during the pre-conception period. Brown et al. [110] also could not reach a conclusion as to a possible association between SRI in pregnancy and ASD in the offspring.

Fatima et al. [111] recently assessed the data only of studies reporting on a positive association between prenatal SRI exposure and ASD. The authors concluded that there is some signal of a possible association, but their conclusion is weakened by the fact that they did not include any of the “negative” studies that did not report any association.

In summary: presently there is inadequate evidence for an association between SRI in pregnancy and ASD in the offspring, especially when controlling for possible confounding factors. One of the possible explanations for the differences in the results of the studies may lie in genetic and epigenetic differences among the populations. SRI may have epigenetic effects, and epigenetic changes are known to be associated with ASD [112].

## 7. Epigenetic Changes

In the embryo and fetus serotonin also serves as a growth factor regulating the proliferation, migration, differentiation and connectivity of neurons, thus controlling to a large extent the development of the brain and other nervous tissues [3,8]. Moreover, serotonin is also a modulator of the epigenome having long-term influence on the future behavior and mental status. The SRI, by crossing the placenta and the blood–brain barrier, may increase fetal brain serotonin and modify serotonin signaling, potentially altering behavior at childhood, adolescence and adulthood [3,8]. Hence, theoretically, SRI might alter different aspects of brain function. Serotonin, norepinephrine and dopamine (monoamines) play important roles in the regulation of the hypothalamic, pituitary adrenal axis (HPA axis), and HPA hormones might affect these systems in utero as well as postnatally, playing an important role in stress response. They may also be involved in the pathogenesis of depression and other psychiatric illnesses [3,8,113,114,115,116]. 

Serotonin, by enhancing the secretion of corticotropin releasing factor (CRF, CRH) by the hypothalamus, up-regulates the HPA axis and increases cortisol in the blood, urine and cerebro-spinal fluid. Changes in the brain levels of the glucocorticosteroids as a result of SRI-induced increased serotonin, may in turn induce epigenetic changes [117,118]. This probably stems from altered feedback inhibition by glucocorticosteroids and an exaggerated response to ACTH. 

There are multiple examples of prenatal causes for epigenetic changes. Epigenetic changes have been observed in the offspring of the Cohen strain of diabetic rats and in human placenta [119,120]. Pregestational or gestational diabetes in human may induce the “metabolic syndrome” in the offspring characterized by increased rate of cardiovascular disease, hypertension and diabetes at adulthood [120]. Hence, modulators of serotonin metabolism (i.e., SRI) are natural candidates to induce epigenetic changes in the fetal brain.

DNA methylation modifications have been often associated with depression [121] which, in turn, may induce changes in the HPA axis, increasing the secretion of CRF [117,118], further implying that modifications of fetal serotonin might induce epigenetic changes. There are many studies in mice and rats showing that prenatal or early postnatal administration of SRI induce epigenetic modifications in the HPA axis leading to distinct neurobehavioral changes [122,123,124,125,126]. It is, however, surprising that there are only few human studies which examined this possibility.

Gurnot et al. found [127] that prenatal exposure to SRI increased DNA methylation status of the CYP2E1 gene in the umbilical cord white blood cells of neonates, unrelated to the changes in DNA methylation observed in depressed patients. 

Several human studies have assessed the HPA axis in infants exposed prenatally to SRI. Oberlander et al. [128] studied the function of the HPA axis in three months old infants prenatally exposed to SRI in comparison to non- exposed infants. SRI reduced the post-stress salivary cortisol, demonstrating an altered HPA stress response. Thus, prenatal SRI, as well as antenatal and postnatal maternal mood, affected the HPA system programming via epigenetic changes.

Pawluski et al. [129] also found that prenatal SRI altered serum cortisol and corticosteroid binding globulin (CBG) in neonates of depressed mothers. 

In summary: serotonin is involved in the function of the HPA axis, directly and indirectly, via modulation of glucocorticosteroids. SRI may also induce epigenetic changes in the HPA axis, as demonstrated in several human studies. These effects of SSRIs might explain the increase in several psychiatric diseases in adolescents prenatally exposed to SRIs, as shown in several, relatively large, clinical studies.

## 8. Pregnancy-Induced Changes in SRI Metabolism, Interaction with Genotype, Pharmacodynamics and Pharmacokinetic Changes

SRIs are metabolized and mostly deactivated in the liver by different cytochrome P450 (CYP) enzymes [130,131,132,133,134,135,136]. Only fluoxetine is a prodrug that is metabolized to norfluoxetine—its active metabolite. The more specific enzymes metabolizing SRI are CYP2D6, CYP2C2 and CYP2C19 [130,131,132,133,134,135,136]. Polymorphism to these enzymes can lead to changes in the metabolic clearance of SRI affecting their serum concentrations. A recent dosing guideline for SRI considering CYP2D6 as well as CYP2C19 has been published [137]. However, it is not clear whether the proposed guidelines are relevant to pregnant women due to the flat concentration-response profile of these drugs.

Higher metabolic rate of most SRI in late pregnancy would lead to lower maternal serum concentrations, which could lead to decreased efficacy. For example, venlafaxine concentrations appear to significantly decrease during pregnancy when compared to the post-partum period [138]. The mean norfluoxetine/fluoxetine metabolic ratio increases over two folds late in pregnancy compared to two months postpartum, due to increased demethylation of fluoxetine by CYP2D6, decreasing the serum levels of the active metabolite and possibly leading to therapeutic failure [139]. Hence, pregnant women using the same dose schedule of venlafaxine throughout pregnancy are at risk for sub therapeutic concentrations, therefore routine monitoring of concentrations of venlafaxine is recommendable during pregnancy, especially in women who experience worsening of their depression. 

Using a physiologically-based pharmacokinetic (PBPK) model to evaluate the need to change dose levels in late pregnancy, the model predicted mean paroxetine steady-state plasma concentration (Css) ratio (postpartum: third trimester) to be 7.1 as compared to the observed value of 3.7 [140]. Sensitivity analysis suggested that a 100% induction of CYP2D6 during the third trimester was required to explain the observed postpartum: third trimester ratios of paroxetine concentrations. Based on these data, the magnitude of late pregnancy hepatic induction of CYP2D6 ranges from 100% to 200%.

In one study, pregnant women on paroxetine during pregnancy [141] who were extensive (EMs) or ultra-rapid metabolizers (UMs) for CYP2D6 exhibited steadily decreasing plasma levels during the course of pregnancy. In contrast, levels of intermediate (IMs) and poor metabolizers (PMs) increased during pregnancy. Decreasing plasma concentrations in EMs and UM are consistent with induction of CYP2D6 activity during pregnancy. Accumulation of paroxetine in women with impaired CYP2D6 metabolism must be cautiously considered because only few women with this genotype were followed up. Of clinical relevance, among the EMs/UMs cases, the depressive symptoms increased significantly during the course of pregnancy, while in the IM/PM group symptoms did not change. This would translate to a need to increase the drug’s dose to account for the lower levels. Berard and colleagues [142] studied the association between the *CYP2D6* genotype and the likelihood of antidepressant discontinuation or dosage modification and found that the likelihood of discontinuing antidepressants during pregnancy was four times higher among slow compared to those with a faster metabolism rate. Thus, prior knowledge of the *CYP2D6* genotype may help in identifying pregnant women who are more likely to discontinue antidepressants. However, this suggestion is not supported by the data itself as women may discontinue SRI for reasons unrelated to drug levels. Indeed, in the study itself, the *CYP2D6* genotype did not appear to affect the dose of SRI taken by the women. The study, despite large numbers, shows unsettled inconsistencies: Genotype did not appear to affect the dose, but did appear to be associated with clinical depression, such that more rapid metabolism could have resulted in lower levels and higher likelihood of depression. 

There is almost no published data on pregnancy-induced pharmacodynamics changes in SRI. Similarly, little data exists in in animals regarding pharmacokinetic and pharmacodynamic changes during the different phases of pregnancy [143]. A systematic review has documented that maternal depression is more prevalent during the first trimester of pregnancy than later on when there has not been yet changes in pre-pregnancy SRI metabolism [144]. This has been partially explained by the high rates of nausea and vomiting of pregnancy (NVP) typical of the first trimester, with its characteristic poor quality of life and foul mood. It is also possible that, once realizing they have conceived, many women abruptly discontinue their SRI in fear of birth defects.

Infants exposed in utero to SRI show up to 30% rates of discontinuation syndrome, which has been described earlier. In adults, such effects have been related the serotonin transporter (SLC6A4) promoter genotype [145]. Reduced 5 min Apgar scores were partially moderated by the sickle Cell anemia ss genotype and neuromotor symptoms were increased while the risk for respiratory distress increased with the ll genotype [146]. Presently, current knowledge does not support pharmacodynamics changes in SRI during pregnancy, and the clinician needs to follow women carefully for increased symptoms which may necessitate increased dosing.

In summary: pregnancy may increase the metabolic inactivation of SRI, necessitating the administration of higher doses, especially during the third trimester of pregnancy and readjustment after delivery. It is unknown whether the different genotypes of CYPD2D6 or other cytochrome P450 enzymes modify the effectiveness of SRI treatment during pregnancy. I addition, there is practically no data regarding pharmacodynamics and pharmacokinetic changes in SRI metabolism in human pregnancy, and how this might affect the clinical response to treatment, if at all. The animal data is also very scant. In any case, the treating physician should monitor the pregnant woman and readjust the dose according to the clinical needs. Additional studies are needed to assess these issues which might be of clinical importance.

## 9. Future Research Directions and Conclusions

Much of the data described here is inconclusive and the final interpretation is difficult, as the possible effects of the underlying disease and the severity of the symptoms were not adequately ascertained. Even the more recent, large population-based studies that compare the data on SRI with that obtained from non-treated depressed mothers are generally based on prescriptions and women might not have taken the drug or not use the dose prescribed. Only large cohort studies taking into account all these weaknesses might reach more reliable and conclusive answers. 

In addition, the numerous studies demonstrating distinct neurobehavioral, epigenetic and other damaging effects of SRI in rodents may point to similar effects in human. The studies demonstrating increased rate of psychiatric diseases in the offspring of SRI treated mothers, which are explained by heritable factors, may also reflect the epigenetic changes possibly induced by SRI. There is, therefore, a need for additional and appropriate human studies to shed light on this issue. 

Due to the importance of serotonin in embryonic and fetal development, it might be expected that prenatal exposure to SRI will impose some dangers to the conceptus. However, most human studies did not demonstrate significant long-term damage following prenatal exposure to SRI, even in children who demonstrated poor neonatal adaptation. 

In the considerations for pharmacological treatment in pregnancy we have to consider the benefit to the mother during pregnancy and thereafter and create the optimal home environmental for the child. Hence, the need for treatment continuation should be carefully considered and discussed with the pregnant mother.

## Figures and Tables

**Table 1 ijms-20-02370-t001:** Summary of SRI effects in pregnancy.

Issue Studied	References	Results	Comments
Original studies looking at major congenital malformations	[13,14,15,16,17,18,19,20]	No increase	Consensus in most studies that, in general, there is no increase in the rate of major malformations
Original studies looking at cardiac malformations	[15,21,22,23,24,25,26,28,29,30,31,32,33,34]	Possible increase of severe cardiac anomalies	Issue in debate: Large studies are generally positive, especially demonstrating an increase in rare, severe, cardiac malformations. Other studies have shown a similar increase in untreated mothers with depression
Original studies looking at miscarriage and stillbirths	[15,47,48,49,50,51,52]	Generally no increase	The slight increase is related to maternal disease or gestational age at study
Original studies looking at preterm birth and low birth weight	[49,52,53,54,55,56,57,58,59]	Generally no increase in preterm birth and no effects on fetal growth	The slight increase in preterm birth and possible decreased fetal growth are related to maternal disease

**Table 2 ijms-20-02370-t002:** Summary of effects of serotonin reuptake inhibitors (SRI) in pregnancy: postnatal effects.

Issue Studied	Reference	Results	Comments
Poor neonatal adaptation	[63,64,65,66,67]	SRI may interfere with neonatal adaptation	Present in up to 30% of newborn infants. Generally no long-term sequelae
PPHN	[6,68,69,70,71,72]	SRI may increase the rate of PPHN	Less than 1%, not very severe, apparently no death reported

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
