# Peer review of "SSRIs and SNRIs (SRI) in Pregnancy: Effects on the Course of Pregnancy and the Offspring: How Far Are We from Having All the Answers?"

_ijms, 2019, doi:10.3390/ijms20102370_

Round 1
Reviewer 1 Report
The authors carried out a detailed and critical analysis of the results of studies on the effects of SSRIs and SNRIs on prenatal development and neonatal health. This subject is very important, because of the importance of serotonin in embryonic and fetal development. The work is valuable and will probably be of great interest.
Author Response
We are grateful for the warm words.
Reviewer 2 Report
Manuscript ijms-493733, "SSRIs and SNRIs (SRI) in pregnancy: effects on the course of pregnancy and the offspring: how far are we from having all answers?" by Ornoy and Koren.
In this manuscript, the authors are summarizing the effects of prenatal exposure to serotonin selective reuptake inhibitors or to serotonin norepinephrine reuptake inhibitors on embryonic development in humans. This review highlights the slight increase in cardiac malformations, in neonatal behavioral changes (poor neonatal adaptation) and in persistent pulmonary hypertension in newborn following prenatal exposure to serotonin selective reuptake inhibitors. The authors then discuss about the transient motor delay and language impairment and the lack of association with autism spectrum disorder of prenatal exposure to serotonin selective reuptake inhibitors. Their putative effect on hypothalamic hypophysis adrenal axis and in particular the associated epigenetic changes are also discussed. The contribution of changes in metabolism of these compounds during late pregnancy may vary in different metabolizers. The authors conclude that the benefits to the mother should be balanced with the risk to the developing embryo/fetus.
This review is fairly clear and well written. Only few modifications would improve this manuscript.
-It is important to distinguish and discuss putative effects of individual molecules not SSRI vs SNRI, or even SRI, fluoxetine may for example have distinct effect from citalopram. Dose dependent effect should also be discussed if available since again effect may vary with the dose.
-This review is focused on developmental and behavioral outcome, but several peripheral actions of SSRIs are not even mentioned:
Thrombose and hemostasis, vasoconstriction, gut motility, hematopoiesis/immunology, placenta direct effect (SERT is highly expressed by trophoblast cells).
-The importance of serotonin action on placenta is missing and putative deleterious effect of blocking placenta serotonin transporter should be discuss including pre-eclampsia, see for example Bottalico et al., Placenta 2004 vol. 25 pp. 518-29; Muller, et al., Neuropsychopharmacology, 2017 vol. 42 pp. 427-436; Laurent et al., Birth Defects Res Part A Clin Mol Teratol, 2016 vol. 106 pp. 1044-1055).
-The introduction should be corrected: "Serotonergic neurons are found in many areas of the cerebral hemispheres, including the prefrontal cortex, hippocampus, nucleus accumbens, corpus striatum; serotonin is present in the entero-chromaffin cells of the gastrointestinal system and in platelets [1,2]." should be modified as "Serotonergic neurons are restricted to brain stem raphe nuclei projecting in many areas of the cerebral hemispheres, including the prefrontal cortex, hippocampus, nucleus accumbens, corpus striatum; in periphery, serotonin is synthesized by the entero-chromaffin cells of the gastrointestinal system and stored in platelets [1,2]."
-Paragraph 8-9-10 should be fused since they are about complementary issues. Several studies on animal models are not included but would complete this discussion e.g. Velasquez et al., ACS Chemical Neuroscience 2016 vol. 7 pp. 327-38).
-Reference 4 is missing from the reference list and in reference 20 2 authors are underlined without reason.
Author Response
We thank the reviewers for their important comments which will improve this review.
Reviewer 2:
The purpose of this review was to discuss only the teratogenic and neuro-teratogenic effects of SRI and SNRI in human pregnancy. We therefore avoided any discussion of animal studies because of the vast literature and the difficulties of choosing the more important studies that fit the purpose of this review. In that line, we do not think that discussing the different important studies, rightly suggested by this reviewer, will serve our purpose. Hence, we could therefore address only part of this reviewer's comments.
"It is important to distinguish and discuss putative effects of individual molecules not SSRI vs SNRI, or even SRI, fluoxetine may for example have distinct effect from citalopram. Dose dependent effect should also be discussed if available since again effect may vary with the dose".
We purposefully tried not to address individual drugs because we think that it will expand the manuscript more than we planned initially.
This review is focused on developmental and behavioral outcome, but several peripheral actions of SSRIs are not even mentioned:
Thrombose and hemostasis, vasoconstriction, gut motility, hematopoiesis/immunology, placenta direct effect (SERT is highly expressed by trophoblast cells).
-The importance of serotonin action on placenta is missing and putative deleterious effect of blocking placenta serotonin transporter should be discuss including pre-eclampsia, see for example Bottalico et al., Placenta 2004 vol. 25 pp. 518-29; Muller, et al., Neuropsychopharmacology, 2017 vol. 42 pp. 427-436; Laurent et al., Birth Defects Res Part A Clin Mol Teratol, 2016 vol. 106 pp. 1044-1055).
These citations are of studies carried out in rodents, and this is beyond the scope of this review. We added a sentence regarding the serotonin transported in human placenta (a study by Ramamoorthy et al, reference number 11), but prefer not to expand further.
-The introduction should be corrected: "Serotonergic neurons are found in many areas of the cerebral hemispheres, including the prefrontal cortex, hippocampus, nucleus accumbens, corpus striatum; serotonin is present in the entero-chromaffin cells of the gastrointestinal system and in platelets [1,2]." should be modified as "Serotonergic neurons are restricted to brain stem raphe nuclei projecting in many areas of the cerebral hemispheres, including the prefrontal cortex, hippocampus, nucleus accumbens, corpus striatum; in periphery, serotonin is synthesized by the entero-chromaffin cells of the gastrointestinal system and stored in platelets [1,2]."
We thank the reviewer for his suggestion. This was modified as suggested
-Paragraph 8-9-10 should be fused since they are about complementary issues. Several studies on animal models are not included but would complete this discussion e.g. Velasquez et al., ACS Chemical Neuroscience 2016 vol. 7 pp. 327-38).
These paragraphs were combined with only one summary at the end. The citation of the paper by Velasquez et al was added just to show that there are also only few animal studies.
-Reference 4 is missing from the reference list and in reference 20 2 authors are underlined without reason.
Reference 4 is there. We removed any unnecessary underlying line.

Reviewer 3 Report
The manuscript proposed by Ornoy and Koren provides an overview about the use of SSRI and SNRI during pregnancy. The review is quite interesting and I have just few minor concerns:
- it would be useful and interesting if the auhtors could provide a table in which they summarize the provided studied with refernces, SSRI and SNRI used, side effect reported, if are meta-analysis or not etc.
-lines 36-38: please provide specific references for each brain regions
Author Response
We thank the reviewers for their important comments which will improve this review.
Reviewer 3:
it would be useful and interesting if the authors could provide a table in which they summarize the provided studied with references, SSRI and SNRI used, side effect reported, if are meta-analysis or not etc.
We thank for this suggestion. We added one table on the effects of SRI during pregnancy and another one on postnatal effects. We cited, however, only original studies as meta-analyses may be sometimes confusing.
-lines 36-38: please provide specific references for each brain regions
Done. The references cited deal with serotonin in the different areas of the brain. We replaced reference 2 by another more detailed study.